# A Novel Fast MPPT Strategy for High Efficiency PV Battery Chargers

**Jose Miguel Espi** * and **Jaime Castello**

Department of Electrical Engineering, University of Valencia, Avd. Universitat S/N,
46100 Burjassot-Valencia, Spain; jaime.castello@uv.es
* Correspondence: jose.m.espi@uv.es; Tel.: +34-963-543-450

**Abstract:** The paper presents a new maximum power point tracking (MPPT) method for photovoltaic (PV) battery chargers. It consists of adding a low frequency modulation to the duty-cycle and then multiplying the ac components of the panel voltage and power. The obtained parameter, proportional to the conductance error, is used as a gain for the integral action in the charging current control. The resulting maximum power point (MPP) is very still, since the integral gain tends to zero at the MPP, yielding PV efficiencies above 99%. Nevertheless, when the operating point is not the MPP, the integral gain is large enough to provide a fast convergence to the MPP. Furthermore, a fast power regulation on the right side of the MPP is achieved in case the demanded power is lower than the available maximum PV power. In addition, the MPPT is compatible with the control of a parallel arrangement of converters by means of a droop law. The MPPT algorithm gives an averaged duty-cycle, and the droop compensation allows duty-cycles to be distributed to all active converters to control their currents individually. Moreover, the droop strategy allows activation and deactivation of converters without affecting the MPP and battery charging operation. The proposed control has been assayed in a battery charger formed by three step-down converters in parallel using synchronous rectification, and is solved in a microcontroller at a sampling frequency of 4 kHz. Experimental results show that, in the worst case, the MPPT converges in 50 ms against irradiance changes and in 100 ms in case of power reference changes.

**Keywords:** photovoltaic (PV); maximum power point (MPP); maximum power point tracking (MPPT); perturbe and observe (P&O); incremental conductance (IC)

## 1. Introduction

Photovoltaic (PV) battery chargers are designed to maximize the energy extracted from solar panels. This requires the maximization of both electronic and PV efficiencies. The electronic efficiency is increased by a parallel configuration of multiple power converters and a synchronous rectification implemented in each one. Paralleling permits activating/deactivating the converters so that each active converter works near its nominal power, thus saving the conduction and switching losses of the inactive converters. Moreover parallelized solutions allows power scaling and increases reliability. On the other hand, the PV efficiency is defined as the ratio between the average power extracted from the panels and the maximum power that can be extracted at a given irradiance. Maximum power point tracking (MPPT) algorithms automatically adjust the PV voltage at the converter input to get the maximum power for each present irradiance level. When a change in irradiance occurs, an ideal MPPT algorithm should reach the new maximum power point (MPP) as fast as possible, and then remain at the MPP without fluctuations. However, in practice, the MPPT algorithms exhibit oscillations around the MPP and take a certain time to converge, penalizing the PV efficiency.

The MPPT strategies can be classified into two main categories: the stand-alone MPPTs [1–5] and the converter-embedded MPPTs [6–10]. A stand-alone MPPT is an independent module that uses the PV voltage and current to determine the input voltage reference to be transmitted to all converters installed. This is typically implemented using perturb and observe (P&O) algorithms. The advantage of the stand-alone method is that it can be used to manage parallelized converters without having to modify their respective controls. In contrast, a converter-embedded MPPT is programmed in the converter control to determine directly the duty-cycle that maximizes the PV power. It is usually implemented using incremental conductance (IC) algorithms [6–8]. Converter-embedded strategies are much faster than stand-alone strategies, thus presenting a higher PV dynamic efficiency, which makes them more suitable in applications where irradiance changes are fast and frequent [11]. However, the parallel multi-stage arrangement becomes difficult to control using a converter-embedded MPPT, as it calculates a single duty-cycle that maximizes the PV power.

In recent research [6,7], new converter-embedded MPPT strategies based on IC have been presented for a step-up converter that combines a fast convergence with a small fluctuation around the MPP. In [7], the static $g_{dc}$ and dynamic $g_{ac}$ PV conductances are explicitly calculated using a moving average filter (MAF) and a lock-in amplifier (LIA) respectively, and then compared and regulated to be equal using an integrator. The ac components used to calculate $g_{ac}$ are the switching ripple components of the PV voltage and current. The MPPT converges to the MPP in approximately 400 ms. As the method requires a small input capacitance to measure $g_{ac}$, the current ripple is present in the PV current and the MPP fluctuates at the switching frequency. To minimize this effect, a large inductance was utilized to achieve a PV efficiency of 99%. More recently, in [6], the IC is solved in a traditional way by incrementing or decrementing the duty-cycle depending on the sign of the conductance error $g_{ac} - g_{dc}$. The ac components used to evaluate the incremental PV magnitudes are the natural oscillations of the input filter. The MPPT settling time was around 300 ms, and the MPP oscillates at the natural frequency of the filter, resulting in a PV efficiency of 97.5%. In both papers, the high frequency used to calculate the PV AC components makes high frequency sampling rates of above 100 kHz necessary, which increases hardware cost and complexity.

This paper presents a new MPPT strategy for step-down battery chargers that combines the benefits of both stand-alone and converter-embedded methods. The proposed MPPT is integrated with the proportional-integral (PI) current regulator, offering a fast convergence to the MPP in less than 100 ms when the demanded power is higher than available maximum PV power, with smooth transitions to regulation when the required power is lower than the available PV power. The basic idea is to insert the conductance error as an additional gain for the PI's integrator when tracking the MPP. This leads to a fast MPPT but a still MPP with PV efficiency higher than 99%, since the conductance error is null at the MPP. A low frequency modulation of 40 Hz is added to the duty-cycle to get the conductance error by multiplying the AC components of PV power and voltage. As a consequence of the low frequency modulation, the proposed MPPT can be solved at 4 kHz sampling rate by a low-cost microcontroller. Additionally, a droop law is proposed to solve the multiple control of parallel converters, proving that converters' currents can be controlled individually without affecting the MPP operation or battery charge. The proposed MPPT with droop has been assayed in the battery charger shown in Figure 1, where electronic efficiency was improved by means of active rectification using $Q_{R_{j1}}$ and $Q_{R_{j2}}$ transistors and blocking transistors $Q_B$, instead of using Schottky diodes.

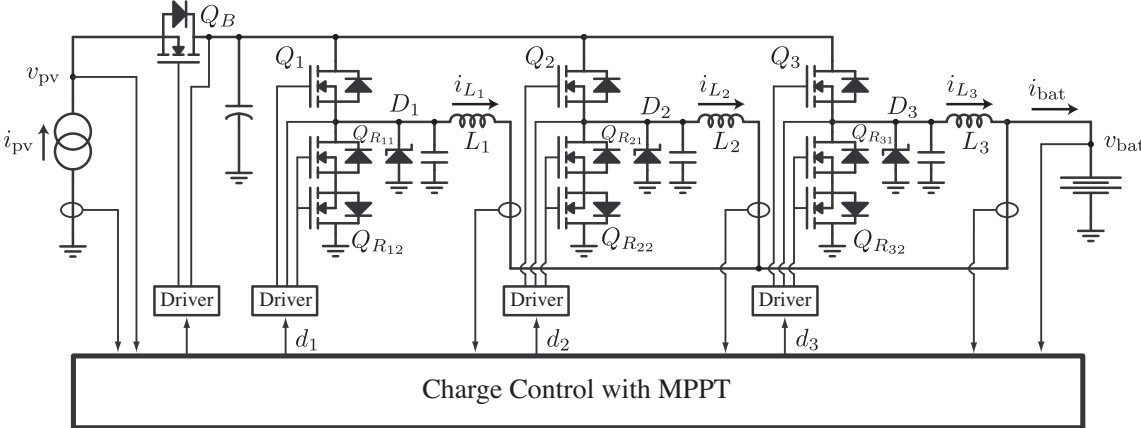

**Figure 1.** Photovoltaic (PV) battery charger using three parallelized step-down converters with synchronous rectification.

## 2. Small-Signal Modeling

Figure 2 shows a single step-down converter with synchronous rectification, thus operating always in continuous conduction mode. The circuit shows averaged values of transistors currents, $d \cdot i_L$ and $(1 - d) \cdot i_L$, where $d$ is the converter duty-cycle and $r$ stands for the inductor series resistance. All relationships can be gathered into the block diagram shown in Figure 3, which constitutes a large-signal model. The function $f_{pv}$ solves the current $i_{pv}$ of the PV panel, using the characteristic I-V curve for a given irradiance and voltage $v_{pv}$. In case of a parallelized step-down converters, $d$ and $i_L$ are vectors containing all duty-cycles and inductor currents, $d \cdot i_L$ is a dot product and $d \cdot v_{pv}$ is a vector. Notice that, if all converters use the same filtering inductance and receive approximately the same duty-cycle, the presented model is valid just considering that $i_L = \sum i_{L_j} \equiv i_{bat}$ is the battery charging current, and $L = L_j/n$, $r = r_j/n$, where $n$ is the number of active parallelized converters, since all active inductors can be considered as operating in parallel.

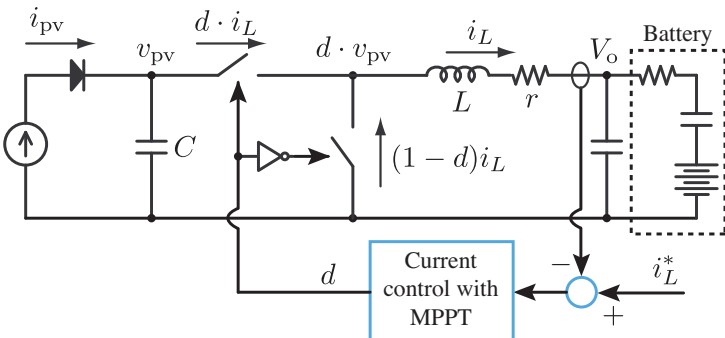

**Figure 2.** Large-signal averaged circuit of a PV step-down converter with active rectification.

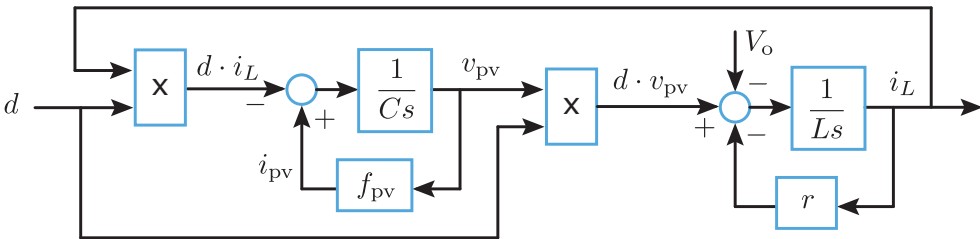

**Figure 3.** Block diagram of the large-signal model.

As the battery voltage $V_o$ changes much slower than all other circuit variables, it can be assumed constant and the small-signal block diagram results as depicted in Figure 4, where the PV panel voltage and current are related through the incremental conductance $g_{ac} = -\frac{di_{pv}}{dv_{pv}}$. From this figure, the duty-to-voltage and duty-to-current small-signal transfer functions are deduced

$$G_v(s) \equiv \frac{\tilde{v}_{pv}}{\tilde{d}}(s) = \frac{G_{cap}(s) \cdot (I_L + DV_{pv}G_{ind}(s))}{1 - D^2 G_{cap}(s)G_{ind}(s)}, \tag{1}$$

$$G_i(s) \equiv \frac{\tilde{i}_L}{\tilde{d}}(s) = \frac{G_{ind}(s) \cdot (V_{pv} + DI_L G_{cap}(s))}{1 - D^2 G_{cap}(s)G_{ind}(s)}, \tag{2}$$

where $G_{cap}(s) = -1/(Cs + g_{ac})$ and $G_{ind}(s) = 1/(Ls + r)$.

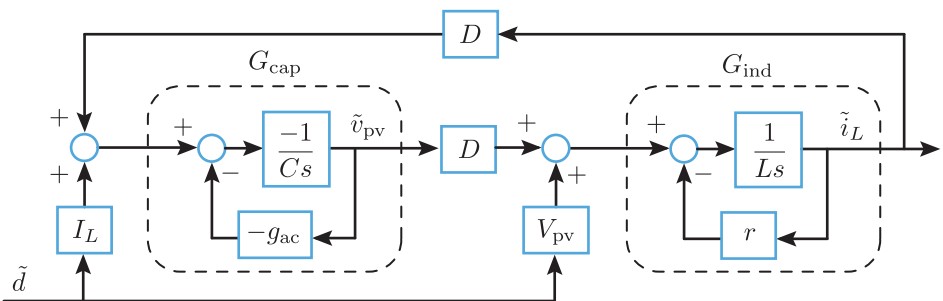

**Figure 4.** Small-signal model of the power converter and PV panel.

Using the steady-state relationships $DV_{pv} = V_o$ and $DI_L = I_{pv}$, basic manipulations reveal that the small-signal transfer functions (1) and (2) can be expressed as

$$G_v(s) = \frac{-k_v(\frac{s}{\omega_{zv}} + 1)}{(\frac{s}{\omega_n})^2 + 2\zeta(\frac{s}{\omega_n}) + 1}, \tag{3}$$

$$G_i(s) = \frac{k_i(\frac{s}{\omega_{zi}} + 1)}{(\frac{s}{\omega_n})^2 + 2\zeta(\frac{s}{\omega_n}) + 1}, \tag{4}$$

where the natural frequency $\omega_n$ and damping factor $\zeta$ are

$$\omega_n = \sqrt{\frac{g_{ac}r + D^2}{LC}}, \tag{5}$$

$$\zeta = \frac{1}{2} \cdot \frac{(r/Z_b + g_{ac}Z_b)}{\sqrt{g_{ac}r + D^2}}, \tag{6}$$

being $Z_b = \sqrt{L/C}$. The frequencies of the zeros $\omega_{zv}$ and $\omega_{zi}$ are

$$\omega_{zv} = \frac{r + DV_o/I_{pv}}{L} \approx \frac{DV_o}{LI_{pv}} = \frac{V_o^2}{P_{pv}L}, \tag{7}$$

$$\omega_{zi} = \frac{g_{ac} - g_{dc}}{C}, \tag{8}$$

where $g_{dc} = \frac{I_{pv}}{V_{pv}}$ is the static conductance, and the dc-gains are

$$k_v = \frac{I_{pv}r/D + V_o}{g_{ac}r + D^2} \approx \frac{V_o}{g_{ac}r + D^2}, \tag{9}$$

$$k_i = \frac{V_{pv}(g_{ac} - g_{dc})}{g_{ac}r + D^2}. \tag{10}$$

Equation (8) indicates that $G_i$ presents a non-minimum phase zero when operating at the left of the MPP, where $g_{ac} < g_{dc}$. In addition, Equation (10) shows that $G_i$ presents null dc-gain when operating exactly at the MPP.

## 3. Working Principles of the Proposed MPPT

This paper proposes to embed an MPPT strategy in a PI current controller, so that it behaves as a normal PI when the power demanded by $i^*_{bat} \equiv i^*_L$ is smaller than the present PV power $p_{pv} = i_{pv}v_{pv}$, that is, when $i^*_{bat} < i_L$, but it opens the current regulation loop and starts MPP tracking when $i^*_{bat} > i_L$.

Regarding the MPPT strategy, the basic idea is to detect the slope of the P–V curve and use it as a gain for the integral action of the PI. This slope is detected by adding a small amplitude modulation $d_m$ to the duty-cycle

$$d_m(t) = d_{m_{pk}} \cdot \cos(\omega_m t), \tag{11}$$

where $\omega_m = 2\pi \cdot 40$ rad/s and $d_{m_{pk}} = 5 \cdot 10^{-3} \equiv 0.5\%$ have been used. According to the duty-to-voltage transfer function $G_v$, this produces a voltage modulation $v_m$ in the PV panel. As the modulation frequency $\omega_m$ is much smaller than $\omega_{z_v}$, it holds that $v_m = -k_v \cdot d_m$. The voltage modulation in turn generates a power modulation $p_m$ as shown in Figure 5.

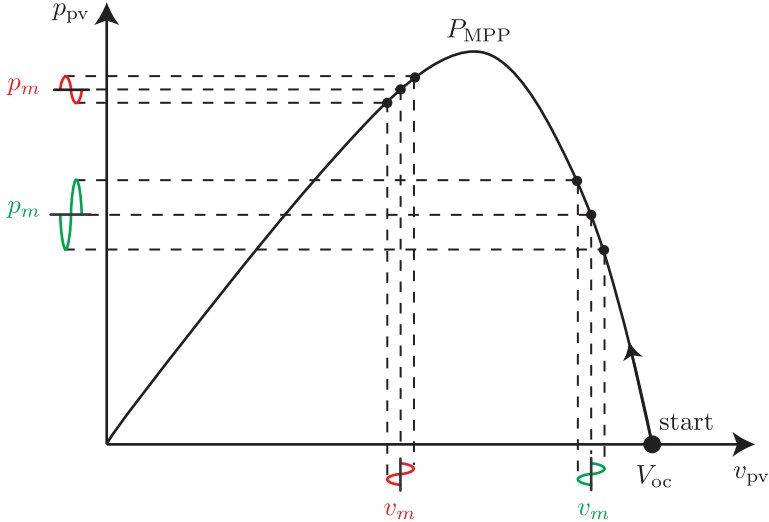

**Figure 5.** Detection of the PV panel operation: to the left of the MPP (red, with $p_m$ and $v_m$ in-phase) and to the right of the MPP (green, with $p_m$ and $v_m$ in anti-phase).

At a given operating point determined by the PV voltage and current levels $(V_{pv}, I_{pv})$, the differential increment of the power is

$$dp_{pv} = I_{pv} \cdot dv_{pv} + V_{pv} \cdot di_{pv} \tag{12}$$

and therefore the power slope is

$$\frac{dp_{pv}}{dv_{pv}} = I_{pv} + V_{pv}\frac{di_{pv}}{dv_{pv}} = V_{pv}(g_{dc} - g_{ac}), \tag{13}$$

which reveals the well known incremental conductance condition $g_{dc} = g_{ac}$ for any local MPP.

In order for the MPPT algorithm to get the value of this slope, a parameter $\delta$ is calculated as

$$\delta(t) \equiv -k_m \cdot p_m(t) \cdot v_m(t). \tag{14}$$

Taking into account that $p_m(t) \approx (\frac{dp_{pv}}{dv_{pv}}) \cdot v_m(t)$ and Equation (13), we get

$$\delta(t) = k_m V_{pv} k_v^2 (g_{ac} - g_{dc}) \cdot d_m^2(t) \tag{15}$$

and, using Equation (11),

$$\delta(t) = \frac{1}{2} k_m V_{pv} (k_v d_{m_{pk}})^2 (g_{ac} - g_{dc})(1 + \cos(2\omega_m t)), \tag{16}$$

which can be separated as $\delta(t) = \delta_{dc} + \delta_{ac}(t)$, where $\delta_{ac} = \delta_{dc} \cdot \cos(2\omega_m t)$ and

$$\delta_{dc} = \frac{1}{2} k_m V_{pv} (k_v d_{m_{pk}})^2 (g_{ac} - g_{dc}). \tag{17}$$

The strategy of the proposed MPPT method is to insert $\delta$ as a multiplying factor between the current error $e$ and the PI controller, as shown in Figure 6. When the current error becomes positive, the MPPT is activated by adding the duty-cycle modulation $d_m$, and the error is multiplied by $\delta$. Only the dc value of $\delta$ (17) generates a dc value for the PI to increase or to decrease the duty-cycle. On the left side of the MPP (point 1 in Figure 7a), it holds that $\delta_{dc} < 0$ and therefore the duty-cycle $d$ decreases and $v_{pv}$ increases (Figure 6b). On the right side of the MPP (point 2 in Figure 7a), $\delta_{dc} > 0$ and $v_{pv}$ decrease (Figure 6c). Hence, when the MPPT is activated, the operating point climbs power automatically to a local MPP (yellow point in Figure 7a). The speed of convergence to the MPP is determined by the integrator gain $k_{i_I}$ and the maximum values of $\delta$ and $e$. In the proposed implementation, $\delta$ is constrained to the interval $[-1, 1]$ and the error is upper-limited to 1A.

Since $|\delta| < 2|\delta_{dc}|$, and $\delta_{dc}$ tends to zero as the operating point approaches the MPP, the power at the MPP is quiescent without oscillations, even if the integrator is set for a fast MPPT, resulting in an excellent static efficiency.

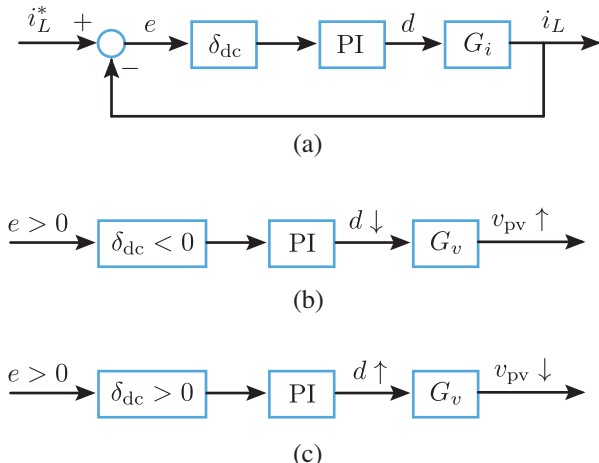

(a)

(b)

(c)

**Figure 6.** Control block diagram (**a**) and MPP tracking situations: (**b**) to the left of the MPP; and (**c**) to the right of the MPP.

When the current error becomes negative due to an increase in irradiance or a decrease in battery power demand (points 3 or 4 in Figure 7b), $d_m$ is set to zero and $\delta$ is set to 1, so the MPPT is transformed into a normal PI current regulation (Figure 6a with $\delta_{dc} = 1$). In this situation, the only possible stable point is on the right side of the MPP (yellow point in Figure 7b).

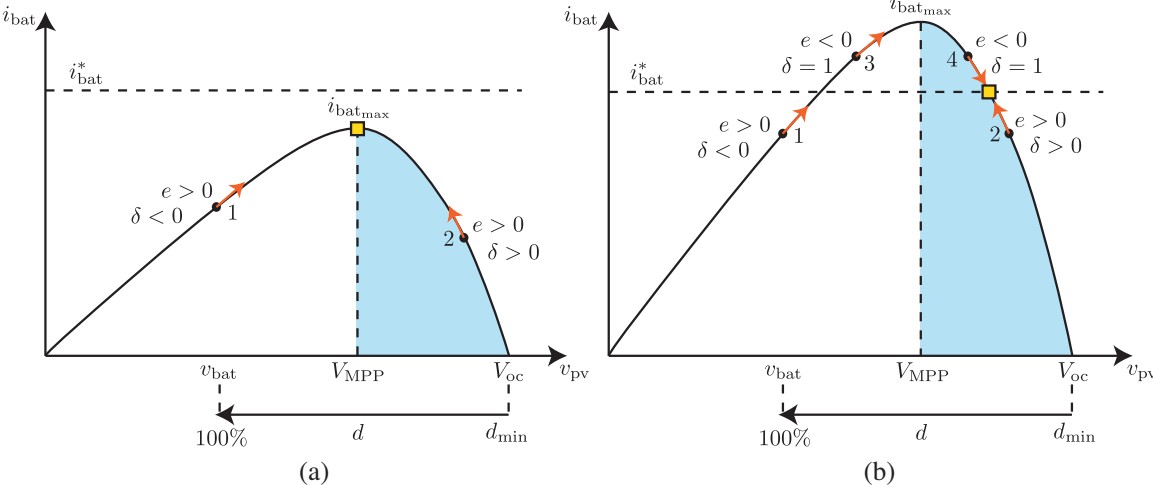

**Figure 7.** Control performance against the two possible scenarios: (**a**) requested power is higher than available PV power; (**b**) requested power is lower than available PV power.

## 4. Description of the Implemented Solution

The proposed strategy has been carried out as depicted in Figures 8–11.

Figure 8 shows the typical PI-based control to regulate the battery voltage. The voltage reference $v_{\text{bat}}^*$ is around 14.7 V/battery during the absorption charge stage, and it is around 13.6 V/battery temperature-compensated float voltage during the float charge stage. The PI determines the charging current $i_{\text{bat}}$, which is limited to $i_{\max} = 60$ A. If the battery SOC is below 80%, this results in a constant current charge at $i_{\max}$ and battery voltage below $v_{\text{bat}}^*$, while at a higher SOC the battery is charged at a constant voltage $v_{\text{bat}}^*$ with current below $i_{\max}$. The voltage reference is changed to the temperature-compensated float voltage when charging current is below $10^{-3}C_{10}$ or absorption time exceeds the 8-hour limit.

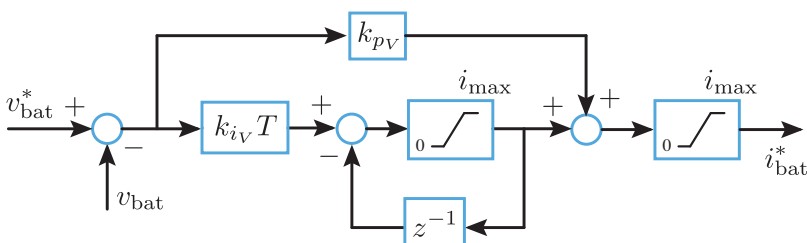

**Figure 8.** Block diagram of the battery voltage regulation to calculate the battery current reference.

Figure 9 illustrates the implemented strategy to detect the converter working point position relative to the MPP. If the current error is positive and the PV current is higher than $i_{\text{start}} = 50$ mA, the MPPT is started by setting MPPT_ON = 1, and $\delta$ is calculated as shown in Equation (14). The voltage $v_m$ and power $p_m$ modulations are extracted from the measured $v_{\text{pv}}$ and $p_{\text{pv}}$, respectively, by means of band-pass digital filters $G_{\text{BP}}(z)$. These filters were implemented using second-order all-pass filters $G_{\text{AP}}(z)$ as

$$G_{\text{BP}}(z) = \frac{1}{2}\left[1 - G_{\text{AP}}(z)\right] \tag{18}$$

and

$$G_{\text{AP}}(z) = \frac{k_2 z^2 + k_1(1 + k_2)z + 1}{z^2 + k_1(1 + k_2)z + k_2}, \tag{19}$$

whose coefficients are calculated as

$$k_1 = -\cos(\omega_0 T),$$
$$k_2 = \frac{1 - \tan(\omega_{\mathrm{BW}} T/2)}{1 + \tan(\omega_{\mathrm{BW}} T/2)},$$

(20)

for a given center-frequency $\omega_0$, bandwidth frequency $\omega_{\mathrm{BW}}$ and sampling period $T = 1/f_s$. Using $\omega_0 = \omega_m$, $\omega_{\mathrm{BW}} = 2\pi \cdot 80$ rad/s and $f_s = 4$ kHz, the programmed all-pass filter resulted in

$$G_{\mathrm{AP}}(z) = \frac{0.8816\, z^2 - 1.8779\, z + 1}{z^2 - 1.8779\, z + 0.8816}.$$

(21)

Gains $k_{pm}$ and $k_{vm}$ must fit the power and voltage oscillations to the interval $[-1,1]$, resulting $k_m \equiv k_{pm} \cdot k_{vm}$ in Equation (14). Values $k_{pm} = 0.5$ and $k_{vm} = 2$ are used to set a high $\delta$ sensitivity, while ensuring the non-saturation of $\delta$ when operating at the neighbourhoods of the MPP.

The condition $i_{\mathrm{pv}} \leq i_{\mathrm{start}}$ in Figure 9 inhibits the MPPT at the start-up, where the duty-cycle is small and operation is in open-circuit with $i_{\mathrm{pv}} = 0$, and hence without any chance to get information by power modulation. Thus, the converter starts on the right side of the MPP with $\delta = 1$, i.e., with a conventional PI action.

On the other hand, when the current error becomes negative, MPPT_ON is set to zero and $\delta = 1$.

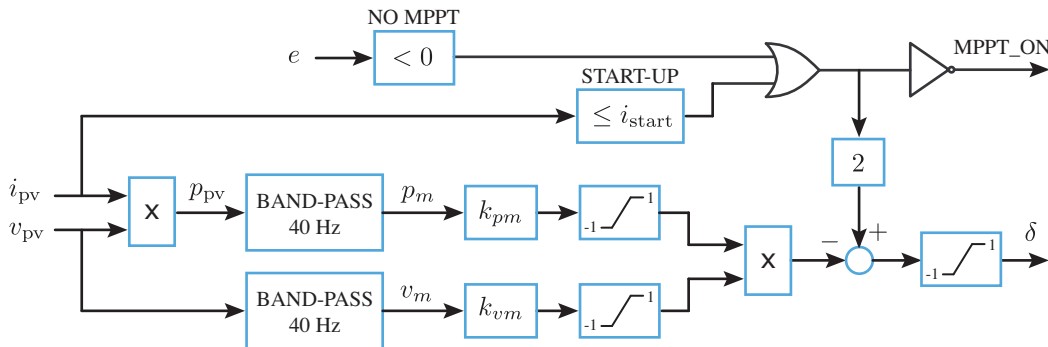

**Figure 9.** Detection of the operating point position relative to the MPP using the parameter $\delta$.

Figure 10 shows the proposed modified PI current control including MPPT action. As mentioned, if $e < 0$, then $\delta = 1$ is applied, the duty-cycle modulation stops, and thus the current control is a conventional PI control. On the contrary, when $e > 0$, the duty-cycle modulation is initiated and $\delta$ is calculated. The current error $e$ is limited to 1 A, so that the speed of convergence to the MPP is given by the integrator gain $k_{i_I}$ and the value of $\delta$, but it is not dependent on the current reference level. As the converter approaches the MPP, $\delta$ tends to zero and the PI slows down the duty-cycle variation to finally get a still MPP operation.

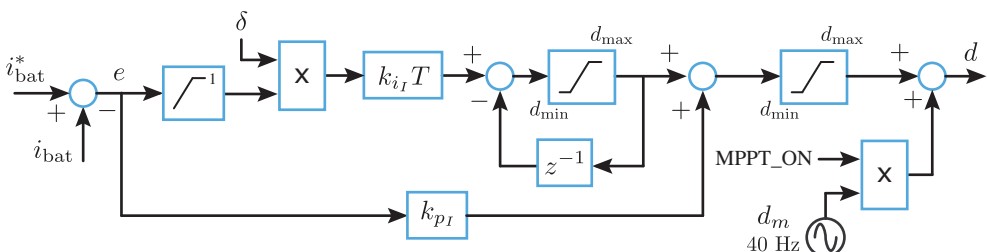

**Figure 10.** Proposed proportional-integral (PI) current control modification to achieve In-Cond MPPT.

The proposed modified PI has to ensure the stability and fast response of the control shown in Figure 6a for all operating points on the right side of the MPP. The design worst case is with $\delta_{dc} = 1$, that is, when MPPT action is inhibited and the controller behaves as a PI compensator. Expressing the PI in its continuous form

$$\text{PI}(s) = k_{p_I} + \frac{k_{i_I}}{s} = k_{p_I} \cdot \frac{s + \omega_z}{s}, \tag{22}$$

the zero $\omega_z$ is designed at the minimum value of the natural frequency

$$\omega_z = \omega_{n_{\min}} \approx \frac{V_o}{V_{\text{MPP}}\sqrt{LC}} \tag{23}$$

in order to get the maximum phase margin as possible.

The gain $k_{p_I}$ is designed to achieve a high control bandwidth by setting the loop-gain crossover frequency $\omega_c$ at $\omega_{\text{nyq}}/6 = \pi/(6T)$. At these high frequencies, the duty-to-current transfer function (4) can be approximated by

$$G_i(s) \approx \frac{k_i \omega_n^2}{\omega_{z_i} s} = \frac{V_{\text{pv}}}{Ls} \tag{24}$$

that exhibits the maximum gain (worst case) when operating at the open-circuit voltage $V_{\text{pv}} = V_{\text{oc}}$ and with all three converters working in parallel $L = L_j/3$. Hence, a simple design equation for $k_{p_I}$ yields $1 = |\text{PI}(j\omega_c)| \cdot V_{\text{oc}}/(L\omega_c)$, or

$$k_{p_I} = \frac{L\omega_c^2}{V_{\text{oc}} \cdot |j\omega_c + \omega_z|} \tag{25}$$

and

$$k_{i_I} = k_{p_I} \cdot \omega_z \tag{26}$$

The designed values for $k_{p_I}$ and $k_{i_I}$ are given in Table 1.

**Table 1.** Control parameters.

| Description | Variable | Value |
|---|---|---|
| Control sampling frequency | $f_s = \frac{1}{T}$ | 4 kHz |
| Duty-cycle modulation frequency | $f_m$ | 40 Hz |
| Duty-cycle modulation amplitude | $d_{m_{\text{pk}}}$ | 0.5% |
| Absorption voltage reference | $v_{\text{bat}}^*$ | 14.7 V/battery |
| Float voltage reference | $v_{\text{bat}}^*$ | 13.6 V/battery |
| Proportional gain - voltage loop | $k_{p_V}$ | 0.05 |
| Integral gain - voltage loop | $k_{i_V}$ | 50 rad/s |
| Current limit | $i_{\max}$ | 60 A |
| Proportional gain - current loop | $k_{p_I}$ | 0.001 |
| Integral gain - current loop | $k_{i_I}$ | 1 rad/s |
| Minimum PV current for MPPT start-up | $i_{\text{start}}$ | 50 mA |
| AC-power gain | $k_{pm}$ | 0.5 |
| AC-voltage gain | $k_{vm}$ | 2 |
| Bandpass filters, center frequency | $f_0$ | 40 Hz |
| Bandpass filters, bandwidth | $f_{\text{BW}}$ | 80 Hz |

Figure 12 shows the resulting Bode diagrams of the open-loop gain at the two ending points of the stable region ($V_{\text{pv}} = V_{\text{MPP}}$ and $V_{\text{pv}} = V_{\text{oc}}$), where the gain crossover frequency $\omega_c$ and both phase and gain margins are detailed. In Figure 12a, the converter operates at the open-circuit voltage with a control bandwidth $\omega_c = 973$ rad/s that ensures a fast response. However, when the converter reaches the MPP in Figure 12b, the control bandwidth is strongly reduced to $\omega_c = 0.1$ rad/s, much lower than the modulation frequency $\omega_m$, and therefore the MPP operation is not affected by the modulation and remains constant without oscillations. The gain and phase margins are large enough to ensure a robust stability in the whole operating range $V_{\text{MPP}} \leq V_{\text{pv}} \leq V_{\text{oc}}$.

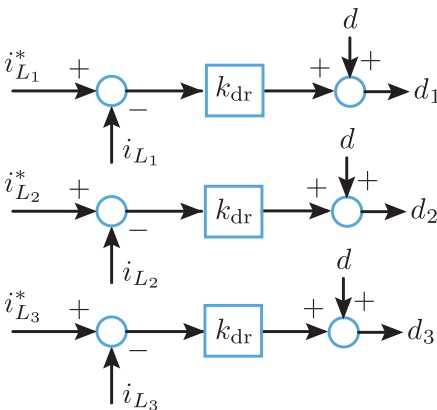

**Figure 11.** Droop correction to equalize and control output currents of all converters.

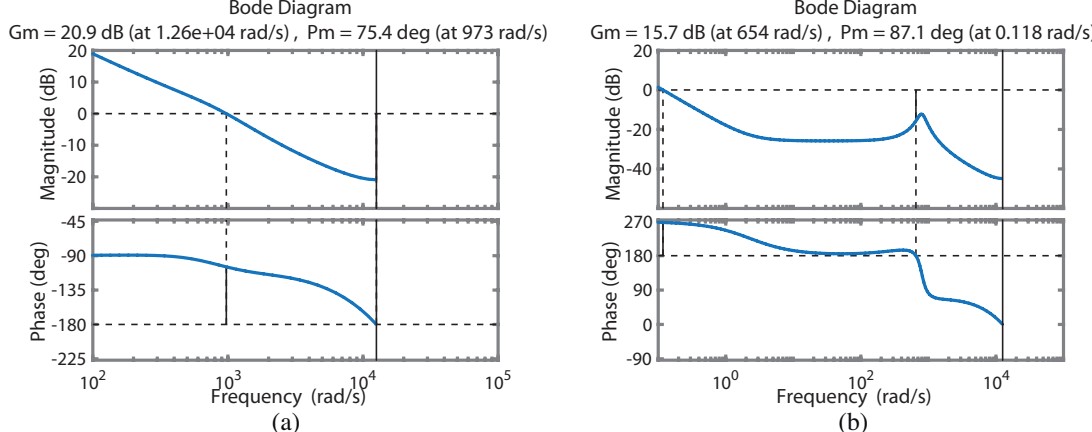

**Figure 12.** Open-loop Bode diagrams when operating at: (**a**) $V_{\text{pv}} = V_{\text{oc}}$; and (**b**) $V_{\text{pv}} = V_{\text{MPP}}$.

Simulated results are presented in Figure 13, obtained using PSIM©, where steps in the charging current reference are applied from 0 A to 20 A. The converter moves from the open-circuit voltage to the MPP in less than 100 ms. The calculated PV efficiency is $\eta = 100 \cdot 422.6/422.9 = 99.9\%$. The ac components $p_m$ and $v_m$ extracted by the band-pass filters and the parameter $\delta$ are also shown. The power $p_m$ oscillates at frequency $\omega_m$ when operating out of the MPP, but at $2\omega_m$ when operation is at the MPP.

Despite a duty-cycle, $d$ is calculated to maximize the extracted PV power if needed, it is not advisable to directly apply it to all parallelized converters, since they have slight differences in the inductors series resistances $r_j$ and turn-on/off delays, which may cause the currents to unbalance. Instead, a droop strategy is proposed as shown in Figure 11 to distribute duty-cycles $d_j$ to the converters. If the $j$-th converter is active, its duty-cycle is calculated as

$$d_j = d + \Delta d_j \, ; \quad \Delta d_j = k_{dr} \cdot (i_{L_j}^* - i_{L_j}), \tag{27}$$

where $i_{L_j}^* = i_{\text{bat}}/n$ is the current reference, $n$ is the number of active converters and $i_{\text{bat}} = \sum i_{L_k}$. On the contrary, when a converter is not active, all transistors $Q_j$, $Q_{R_{j1}}$ and $Q_{R_{j2}}$ in Figure 1 are switched off and therefore $i_{L_j}$ is zero. It can be noticed that the averaged value of all applied duty-cycles to the active converters is $d$

$$\sum_j^n \Delta d_j = 0 \, ; \quad \sum_j^n d_j = nd. \tag{28}$$

Each active converter current satisfies

$$i_{L_j} = (d_j v_{\text{pv}} - V_{\text{o}}) G_{\text{ind}} \tag{29}$$

and adding for all active converters

$$i_{\text{bat}} = \sum_j^n i_{L_j} = (\sum_j^n d_j v_{\text{pv}} - n V_{\text{o}}) G_{\text{ind}} = (d v_{\text{pv}} - V_{\text{o}}) n G_{\text{ind}}, \tag{30}$$

which shows, as mentioned before, that the parallel configuration behaves as a single stage with an averaged duty-cycle $d$ and with all inductances $G_{\text{ind}}$ in parallel.

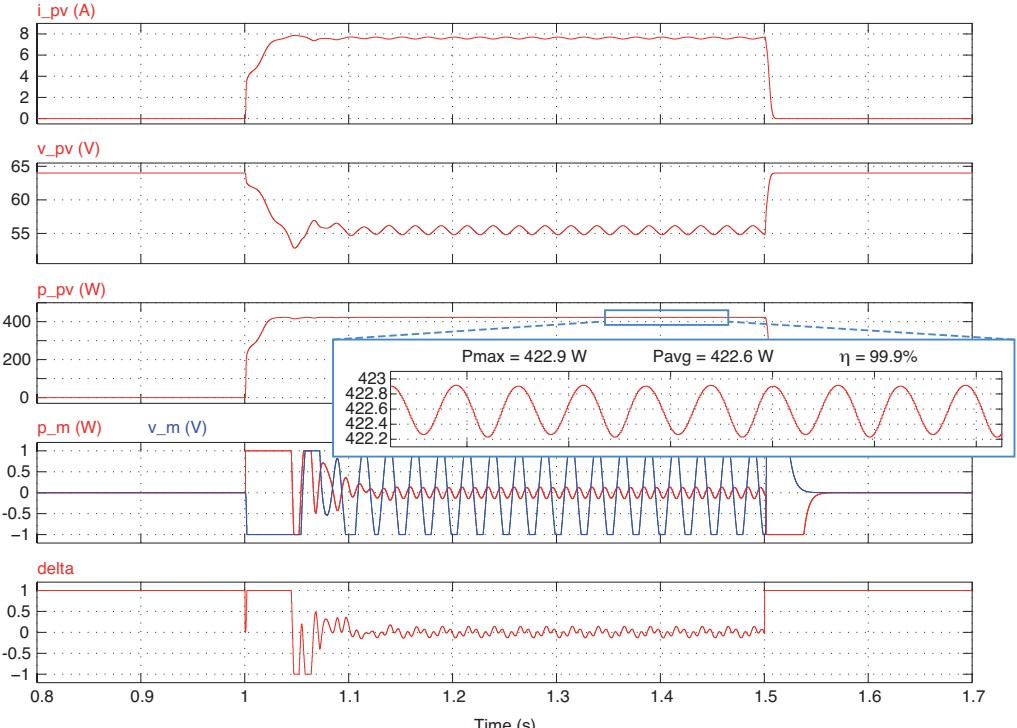

**Figure 13.** Simulated transient response against current reference steps. The power alternates between zero and the MPP—from top to bottom: $i_{\text{pv}}$, $v_{\text{pv}}$, $p_{\text{pv}}$, $p_m$, $v_m$ and $\delta$.

Equation (29) in steady-state results $i_{L_j} = (d_j v_{\text{pv}} - V_{\text{o}})/r_j$, and therefore a variation in the duty-cycle $\Delta d_j$ generates a variation in the converter current $\Delta i_{L_j} = \Delta d_j v_{\text{pv}}/r_j$, which gives an estimation for the droop gain $k_{dr}$ in (27). In order to guarantee the currents compensation, the droop gain is designed as

$$k_{dr} \gtrsim \frac{\max r_j}{V_{\text{MPP}}}. \tag{31}$$

The proposed droop strategy allows for controlling each converter current individually to optimize the overall efficiency. For instance, when the power is lower than one third of the total installed power, only one converter is active and the other two are kept off, so that switching losses are minimized. When power is between one third and two thirds of the total power, two converters are active and share the power from 50% to 100% of their rated power. Finally, when the power is higher than two thirds of the total, all three converters are activated and share the power from 66% to 100% of their rated power. Moreover, a rotation strategy is also implemented to alternate the active and inactive converters to equalize transistors aging and to minimize thermal cycling. It will be shown in the experimental results that the converters' activation and deactivation for losses rotation does not have a transient effect on the MPP operation, and hence it can be done without affecting the photovoltaic efficiency.

## 5. Experimental Results

The presented MPPT strategy with droop was assayed in the battery charger shown in Figure 1, whose main parameters are specified in Table 2. Transistors $Q_j$ ($j$ = 1, 2, 3) are fired at $f_{sw}$ = 40 kHz with complementary drive for the rectification transistors $Q_{R_{j1}}$ and $Q_{R_{j2}}$. Schottky diodes $D_j$ drive only during the PWM dead-time. The anti-series transistors $Q_{R_{j2}}$ impede the conduction of the lossy body-diodes of $Q_{R_{j1}}$ during the dead-time. The blocking transistors $Q_B$ are always on, except if the input voltage gets close to the battery voltage or a if a panel reverse current is detected. The converter was designed to charge lead acid batteries with voltage ranging from 12 V to 48 V and charging current up to 60 A. Though the converter is 3 kW rated, presented experimental results were obtained at a lower power, using the 480 W E4350B solar array simulator from Agilent/Keysight Technologies©.

The control is resolved in a RX630 microcontroller from Renesas Electronics© at 4 kHz, using three independent PWM outputs to drive the three converters, and six analog input channels for the PV voltage and current, the battery voltage and the three output currents. A human interface device (HID) class USB communication, readable with computers, tablets, etc., has also been implemented to send internal data at a speed of 2 kB/s.

**Table 2.** Power converter parameters.

| Description | Variable | Value |
|---|---|---|
| Nominal power | $P$ | 3 kW |
| Output voltage | $V_o$ | 12 V–48 V |
| Maximum output current | $i_{bat}$ | $3 \times 20$ A = 60 A |
| Switching frequency | $f_{sw}$ | 40 kHz |
| Output filter inductances | $L_j$ | 130 μH |
| Inductor series resistances | $r_j$ | 25 mΩ |
| Input capacitance | $C$ | 2300 μF |

Figure 14 shows the P–V irradiance curves programmed in the E4350B to test the performance of the proposed MPPT method. Initially, the converter is turned on with irradiance level corresponding to curve PV1, which is maintained for five seconds. Then, the irradiance is suddenly changed to the curve PV2 and is maintained for another five seconds, and so on with curves PV3 and PV4. Finally, the converter is turned off with irradiance PV4. Red circles indicate the converter operating point motion and were obtained in real time via USB with a 2 ms sampling period. It can be seen that there is only one red circle in the transition between two consecutive MPPs, which indicates that the MPPT takes less than 4 ms to converge when the irradiance decreases.

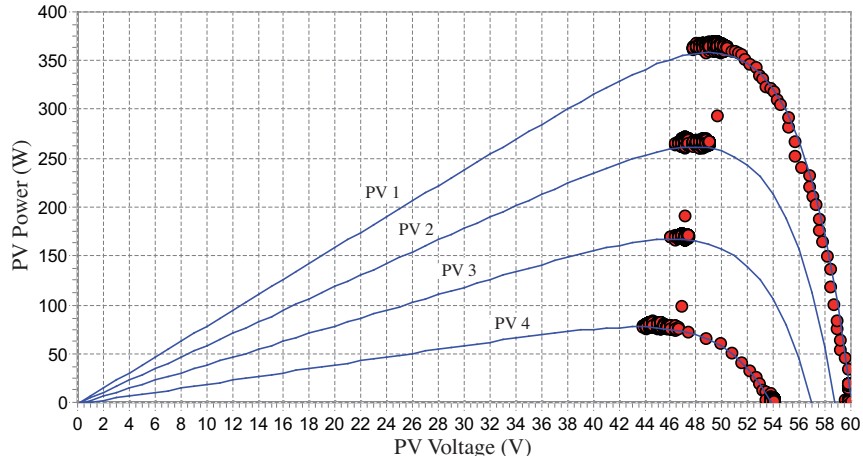

**Figure 14.** Operating point shift during irradiance step changes from PV1 to PV4 (samp. time = 2 ms).

In more detail, Figures 15 and 16 show transients produced by a sudden increase and decrease in irradiance, respectively. In Figure 15, the irradiance is changed from PV4 to PV1. The new MPP is reached in approximately 50 ms. The 40 Hz oscillation is barely distinguishable in the PV current or voltage, and it cannot be observed in the PV power. In Figure 16, the irradiance is decreased from PV1 to PV4, and, as mentioned before, most of the power transition takes less than 4 ms.

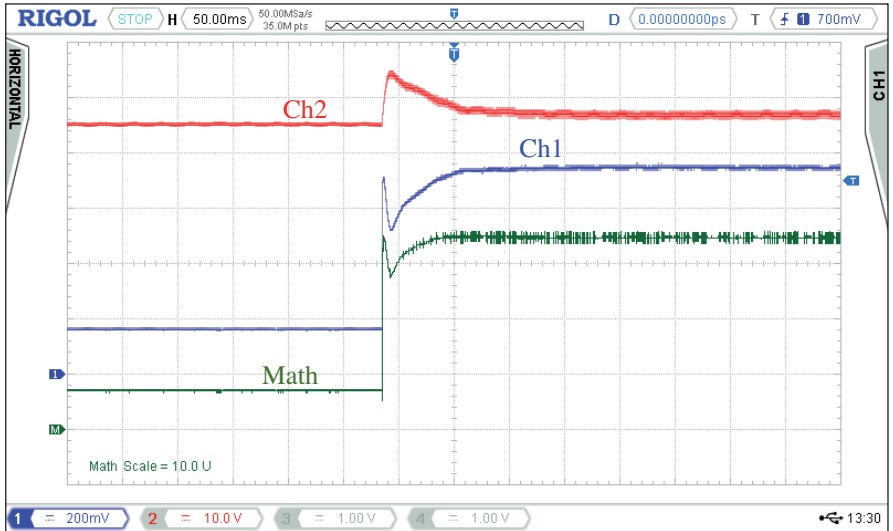

**Figure 15.** Transient response against an irradiance step from PV4 to PV1. Ch1: PV current (2 A/div). Ch2: PV voltage (10 V/div). Math: PV power (100 W/div).

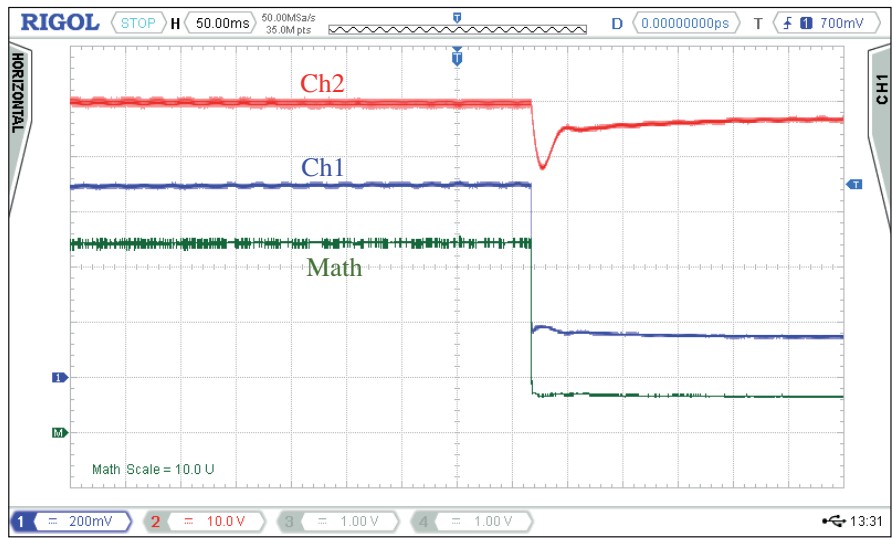

**Figure 16.** Transient response against an irradiance step from PV1 to PV4. Ch1: PV current (2 A/div). Ch2: PV voltage (10 V/div). Math: PV power (100 W/div).

An irradiance change is not the most challenging case in terms of speed response, since the PV voltage and duty-cycle variations are not wide. However, a large step change in the demanded battery charging power requires a significant variation of the duty-cycle and PV voltage, and this is the worst case for the converter settling time. In this sense, Figure 17 shows the transients produced by steps in $i^*_{bat}$ between 0 A and 20 A when the converter operates at the irradiance PV1 curve. The PV power changes from zero to the MPP, showing a rising time of approximately 100 ms and a falling time around 20 ms. It can be noticed again that the power at the MPP is constant without oscillations.

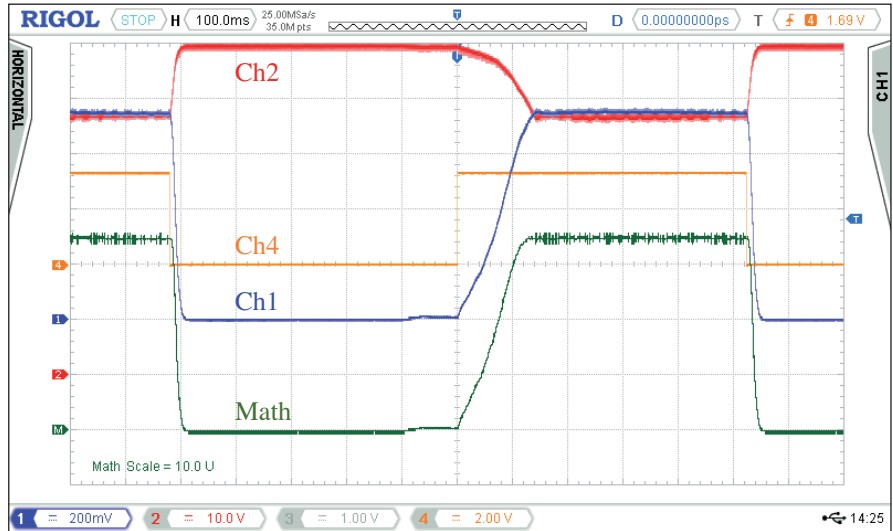

**Figure 17.** Transient response against a current reference $i_{\text{bat}}^*$ steps between 0 A and 20 A. Ch1: PV current (2 A/div). Ch2: PV voltage (10 V/div). CH4: current reference synchronism digital output. Math: PV power (100 W/div).

Finally, Figures 18 and 19 are intended to show the performance of the droop compensation and the on/off switching of parallelized converters. These figures were obtained when charging batteries at 17 A and 28 V giving the maximum 480 W of the solar simulator. At the beginning of Figure 18, the charge current shown in Ch3 is equally shared by converters 1 and 2 by means of droop action. Then, the current reference of converter 1 is set to zero and the reference of converter 2 is set to the total charge current. After around 2 ms, all of the charge current is provided by converter 2, and the converter 1 is turned off. Moreover, Figure 19 shows the effect of a converter activation. Initially, converter 1 is off and all charge currents are provided by converter 2. Then, converter 1 is turned on and current references of both converters are set to half the charge current. Finally, after 2 ms, the current is equally shared by converters 1 and 2. It can be seen that activation and deactivation of converters have no effect on the charging current and hence do not affect the MPP operation.

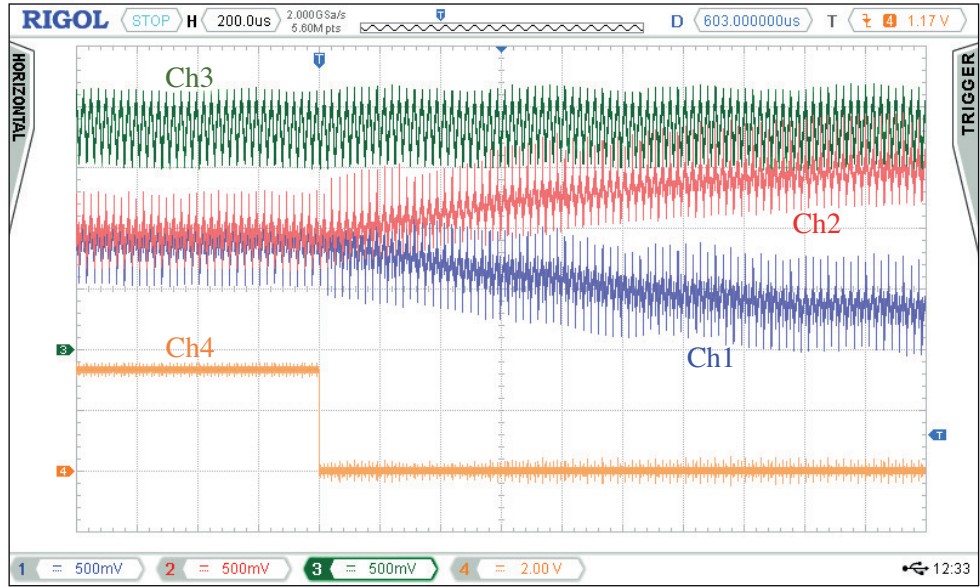

**Figure 18.** Droop equalization and converter 1 shut-down. Ch1: converter 1 output current (5 A/div). Ch2: converter 2 output current (5 A/div). Ch3: total output current (5 A/div). Ch4: synchronism digital output.

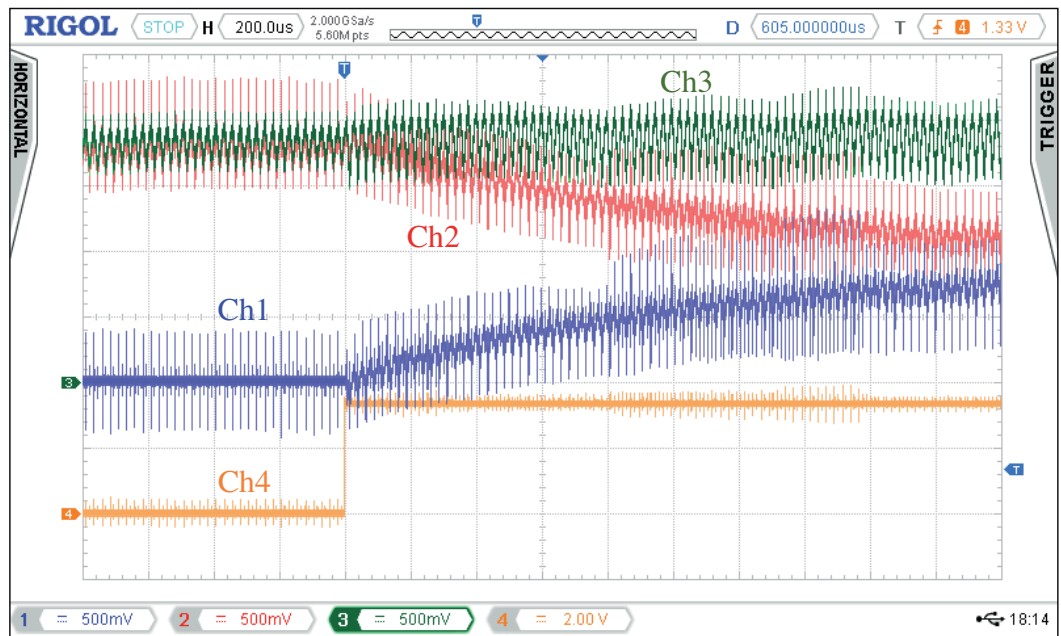

**Figure 19.** Converter 1 turn-on and current sharing with converter 2 using droop correction. Ch1: converter 1 output current (5 A/div). Ch2: converter 2 output current (5 A/div). Ch3: total output current (5 A/div). Ch4: synchronism digital output.

## 6. Conclusions

This paper presents a new fast MPPT method for step-down photovoltaic (PV) battery chargers. The method adds a low frequency modulation to the duty-cycle to calculate the conductance error, which is used as a gain in the current loop. Therefore, it can be considered as another MPPT variant of the incremental conductance (IC) type. This produces a quiescent maximum power point (MPP) since the control bandwidth tends to zero at the MPP, yielding PV efficiencies higher than 99%. However, when the operating point is not close to the MPP, the bandwidth of the current control is around 1 krad/s, resulting in a fast convergence to the MPP.

Furthermore, if demanded power is lower than the available maximum PV power, the proposed design ensures a fast regulation on the right side of the MPP. The presented MPPT exhibits, in the worst case, settling times of 50 ms against irradiance changes, and 100 ms against power reference changes.

In addition, the control problem of a parallel arrangement of converters is solved by means of a droop law. The MPPT algorithm gives an averaged duty-cycle for all active converters, and the droop compensation allows duty-cycles to be distributed to all active converters to control their currents individually. Moreover, the droop strategy allows activation and deactivation of converters without affecting the MPP and battery charging operation.

Finally, it is worth noticing that the proposed battery charger control can be solved at low sampling rates using a low-cost microcontroller.

**Author Contributions:** J.M.E. proposed the main idea, performed the theoretical analysis and wrote the paper. All authors contributed to the practical implementation and experimental validation, paper review and editing. All authors have read and approved the final manuscript.

**Funding:** This research was funded by the Centro para el Desarrollo Technological Industrial (CDTI) under the project grant number IDI-20160123.

**Acknowledgments:** We thank the company Dismuntel S.A.L. for its help in building the PV charger prototype.

**Conflicts of Interest:** The authors declare no conflict of interest.

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
