# Peer review of "A Novel Fast MPPT Strategy for High Efficiency PV Battery Chargers"

_energies, doi:10.3390/en12061152_

Round 1

Reviewer 1 Report

I believe this paper is an interesting and impressive work but authors are encouraged to pay more attention to academic writing. Some comments are attached.

Author Response

Reviewer 1:

This is an impressive work presenting a new MPPT method for photovoltaic (PV) battery chargers. However, authors are highly encouraged to improve their writing style. Some comments are listed as follows:

Page 2 Paragraph 2 Line 7: The “an small input” should be “a small input”; Page 2 Paragraph 3 Line 5: The “consists on” should be “consists of”;
Page 2 Paragraph 3 Line 12: The “without affect” should be “without affecting”; Page 5 Part 3 Paragraph 1 Line 1: The “a MPPT” should be “an MPPT”;

Page 5 Part 3 Paragraph 2 Line 2: The “an small input” should be “a small input”;

Page 6 The Paragraph starting with “The strategy of the proposed MPPT method” Line 4: “to increase or decrease” should be “to increase or to decrease”;

Page 6 The Paragraph starting with “Since |δ| < 2|δdc| tends”: This sentence is not clear enough;

Page 6 The Paragraph starting with “When the current error becomes” Line 1: The “because” should be “because of”;

English corrections have been made in the final version of the paper.

Page 10 Figure 12: The authors are suggested to have a clear explanation about the meaning of curve in each sub figure and their method to get the result (i.e. the tracking time is less than 100ms);

Labels in Fig.12 have been changed to match the name of the variables used in the paper. An explanation has been added in the figure caption. The settling time of 0.1s can be better observed in the new figure.

Page 10 Figure 12: The authors are encouraged to give a higher resolution figure and smaller scale of grid in specific region;

Fig.12 has been completely redrawn in vector graphics format. The simulation was performed using PSIM, and the output data has been exported to a CSV file and plotted in Matlab. The region of the power trace that has been zoomed takes intentionally several cycles, in order to probe that the efficiency measured is in steady regime.

Part 6: In practice, Partial Shading Condition (PSC) is a common problem in MPPT. PSC may affect the MPP accuracy even damage the circuit. The authors are encouraged to discuss their strategy for Partial Shading Condition or to point this problem out. The authors are also encouraged to refer the following paper “An NNwC MPPT-based energy supply solution for sensor nodes in buildings and its feasibility study”, which well discussed this topic;

Part 6: The MPPT is a dynamic process related with not only irradiance but also temperature, there could be multiple maximum and the proposed MPPT method may be trapped in a local maximum. The authors are encouraged to discuss or to point out this problem.

The paper presents a very fast power-climbing method that stabilizes in a local MPP. In case of PSC, the method has to be combined with Global MPPT strategies, either based on continuous search or based on MPP predictions from irradiance and temperature real-time measurements. This is beyond the scope of the paper. However, the speed of convergence of the proposed method offers a realistic chance to implement a continuous-search Global MPPT with small impact on PV efficiency, just by changing the duty-cycle stored in the PI’s integral action (value stored in the Z^-1 in Fig.10) and letting the MPPT to complete a new tracking. Initial and final powers and duty-cycles are stored to find the Global MPP. Authors hope to publish these global strategies in a different article.

Reviewer 2 Report

The paper entitled A Novel Fast MPPT Strategy for High Efficiency PV Battery Chargers presents a new maximum power point tracking (MPPT) method for photovoltaic (PV) battery chargers.

My general comments are as follows:

1. The topic of the paper is in line with the topics of the Energies journal.

2. The research is designed appropriately, and the methods used are adequately described.

3. The results are clearly presented and the results support the conclusions.

4. The authors should reduce the 2nd and 3rd section including fundamentals found in text books as well as be more precise in section 4. 

5. The quality of figure 14 should be improved.

6. Please justify your choices for the parameters in Table 2. Also please include a reference of the table in your  text.

In general the novelty and the significance of content of the paper is high. The quality of presentation and the scientific soundness is satisfactory. The overall merit of the study is positive.

Author Response

Reviewer 2:

The paper entitled A Novel Fast MPPT Strategy for High Efficiency PV Battery Chargers presents a new maximum power point tracking (MPPT) method for photovoltaic (PV) battery chargers.

My general comments are as follows:

1. The topic of the paper is in line with the topics of the Energies journal.

2. The research is designed appropriately, and the methods used are adequately described.

3. The results are clearly presented and the results support the conclusions.

4. The authors should reduce the 2nd and 3rd section including fundamentals found in text books as well as be more precise in section 4.

Section 3 describes the strategy adopted for the proposed MPPT. Only Figure 5 and equations (12) and (13) can be found in the basic related literature, but we believe that this information helps to better understand the working principles of the proposal. In the same regard, information given in section 2 is intensively used to justify the design of the controller and droop parameters.

More detailed information is added to section 4 (see answer to question 6).

5. The quality of figure 14 should be improved.

The experiment shown in Fig.14 has been repeated and drawn again in a vector graphics format.

This figure was obtained using software created by the authors to obtain real-time information of the converter through USB in the HID class. This software has been modified to include an option to export data to metafile, so that it can be drawn offline as EMF file.

6. Please justify your choices for the parameters in Table 2. Also please include a reference of the table in your text.

Table 2 was referenced in text in page 9, at the end of the PI design process, below equation (26). The paper gives design equations for the critical parameters kpI (25), kiI (26) and kdr (31). Just follow the design using the power converter values given in Table 1 and you will get the values of these parameters in Table 2.

The amplitude of the duty-cycle modulation dm_pk is adjusted to produce a modulation of around 0.5V in the PV voltage, using (9). The modulation frequency of 40 Hz can be changed depending on the needs, but it is recommended to be below (5) and (7) to ensure that dm is exactly in opposite phase with respect vm (PV voltage). The constants kpm and kvm have the simple mission of fitting the power and voltage 40 Hz oscillations to the interval [-1,1]. Though Table 2 gives the PI parameters of the voltage loop, these have nothing to do with MPPT and in fact this loop is unused to test the MPPT.

The condition ipv<istart in Fig.9 inhibits the MPPT at the start-up, where the duty-cycle is small and operation is in open-circuit with ipv=0, and hence without any chance to get information by power modulation. So, the converter starts at the right side of the MPP with delta=1, i.e. with a conventional PI action.

These comments are added to the new version of the paper.

Reviewer 3 Report

This paper addresses a very important aspect in photovoltaic
systems, namely the determination of the maximum power operation
point.

The proposed solution is very interesting because it can be
embedded inside the current controller and does not require
a dedicated optimization algorithm that generates setpoint
to the voltage/current controller.

I have only few observations related to the paper:

- why authors proposed the use of two anti-series rectification
  transistors? The explanation presented in the article is to
  impede the conduction of the lossy body-diodes of rectification
  transistor during the dead-time. My opinion is that even if only
  one transistor is used, the Schottky diode will take the current
  from internal body-diode due its lower forward voltage and even
  if this is not happening (in case of high voltage Schottky diodes
  that have forward voltages similar to body-diodes), the short
  duration of the dead-time would make the body-diode losses
  negligible.

- the low amplitude of the duty-cycle modulation leads to a very
  small amplitude of current variation (especially when panel
  operates close to MPP). This small AC signal can be easy affected
  by noises - for example 50Hz noise from power grid that has a
  frequency close to the 40Hz modulation signal.
  Is the band-pass filter efficient enough to reject the noises?
  It would be useful to add a frequency characteristic of the
  filter in order to evaluate its performances.

- I recommend to add a small description of the hardware platform
  used for implementation of the control algorithm. If it is based
  on a microcontroller can you specify what kind of microcontroller
  was used? This description will support one of the main characteristic
  of the proposed algorithm, namely the low complexity / low computational
  resources required for implementation.

In conclusion, my recommendation is to accept this paper after minor revisions.

Author Response

Reviewer 3:

This paper addresses a very important aspect in photovoltaic systems, namely the determination of the maximum power operation point.

The proposed solution is very interesting because it can be embedded inside the current controller and does not require a dedicated optimization algorithm that generates set point to the voltage/current controller.

I have only few observations related to the paper:

-       why authors proposed the use of two anti-series rectification transistors? The explanation presented in the article is to impede the conduction of the lossy body-diodes of rectification transistor during the dead-time. My opinion is that even if only one transistor is used, the Schottky diode will take the current from internal body-diode due its lower forward voltage and even if this is not happening (in case of high voltage Schottky diodes that have forward voltages similar to body-diodes), the short duration of the dead-time would make the body-diode losses negligible.

In active rectification, the body-diode losses are not conduction losses but turn-off switching losses. Consider a conventional active rectification at the dead-time situation, where the output current is driven 80%-20% by the Schottky diode and body-diode respectively. When the main transistor is turned on, a reverse voltage is suddenly applied to the diodes. As the Schottky diode has no depletion zone at the junction, it turns off immediately. However the body-diode presents the reverse recovery phenomenon. During the final stage of the reverse recovery process, the input voltage (PV voltage in our setup) is fully applied to the diode, and hence the reverse current involves diode turn-off losses. In addition, this affects the turn-on losses of the main transistor that has just started to drive, as its current is incremented by the reverse recovery current.

The improved active rectification (Fig.1 in the paper) is implemented in some commercial PV battery regulators like in the famous MorningStar series. The added anti-series transistor prevents the mentioned body-diode from driving during the dead-time interval, in which all transistors and body-diodes are now off and only the Schottky diode is on. More precisely, the body-diode turns off when the anti-series transistor is turned off, at the beginning of the dead-time (previously the body-diode was barely driving current because its own MOSFET was on), and the recovery process is now insignificant (small charge storage QRR) and lossless since the reverse voltage is near zero (the Schottky diode is on). Now, when the main transistor is switched on, no reverse current is produced at all since the body-diode was already off. As a consequence, the turn-on losses of the main transistor are also reduced.

It is important to notice that the added anti-series transistor does not block voltage when is off. This permits using MOSFETs with low VDS and extremely low RDS(on), in a very small package. We have used the IRF6619 (2 in parallel) with RDS(on)=2 mOhm, VDSmax=20V and ID=30A in DirectFET package (6 mm x 5 mm). The transistors that form the conventional active rectification are the IRFB4321 (forming groups of 2 in parallel) and the Schottky diode is STPS20170CT.

Authors are aware that this is relevant information, but it is beyond the scope of the paper.

-       the low amplitude of the duty-cycle modulation leads to a very small amplitude of current variation (especially when panel operates close to MPP). This small AC signal can be easy affected by noises - for example 50Hz noise from power grid that has a frequency close to the 40Hz modulation signal. Is the band-pass filter efficient enough to reject the noises? It would be useful to add a frequency characteristic of the filter in order to evaluate its performances.

The modulation frequency of 40 Hz was just a choice to get a resolution of 100 points/cycle at our 4 kHz sampling frequency, but obviously a different combination of frequencies and resolutions can be used. It works nicely as proposed in a battery solar regulator, as there is no grid connection. But we agree that in a grid-connected application (for instance in a PV generator for a 400 V micro-grid using a Boost topology, on which we are working at this moment) different values may be more suitable.

Regarding the frequency response of the band-pass filters, it can be obtained in Matlab from (18) and (21) by typing (see the attached PDF):

GAP=tf([0.8816  -1.8779  1],[1  -1.8779  0.8816],1/4e3);

GBP=(1-GAP)/2;

bode(GBP);

The center frequency is 2*pi*40=251 rad/s.

The selected bandwidth of 80 Hz (twice the center frequency) gives a good trade-off between filtering and speed. The more selective the filter is (the higher the Q-factor), the longer the time to settle.

- I recommend to add a small description of the hardware platform used for implementation of the control algorithm. If it is based on a microcontroller can you specify what kind of microcontroller was used? This description will support one of the main characteristic of the proposed algorithm, namely the low complexity / low computational resources required for implementation.

The control is resolved in a RX630 Renesas microcontroller at 4 kHz, being necessary 3 independent PWM outputs (to control the three converters) and 5 analog input channels (PV voltage and current, and the three converter output currents). The microcontroller implements also an HID class USB communication, readable with computers, tablets, etc., to send internal data at 2 kbytes/sec.

This has been added to the setup description in section 5 in the new paper version.
